# Visualization-enhanced under-oil open microfluidic system for in situ characterization of multi-phase chemical reactions

Qiyuan Chen[1], Hang Zhai[2], David J. Beebe [3,4,5], Chao Li [3] ✉ & Bu Wang [1,2] ✉

Under-oil open microfluidic system, utilizing liquid-liquid boundaries for confinements, offers inherent advantages including clogging-free flow channels, flexible access to samples, and adjustable gas permeation, making it well-suited for studying multi-phase chemical reactions that are challenging for closed microfluidics. However, reports on the novel system have primarily focused on device fabrication and functionality demonstrations within biology, leaving their application in broader chemical analysis underexplored. Here, we present a visualization-enhanced under-oil open microfluidic system for in situ characterization of multi-phase chemical reactions with Raman spectroscopy. The enhanced system utilizes a semi-transparent silicon (Si) nanolayer over the substrate to enhance visualization in both inverted and upright microscope setups while reducing Raman noise from the substrate. We validated the system's chemical stability and capability to monitor gas evolution and gas-liquid reactions in situ. The enhanced under-oil open microfluidic system, integrating Raman spectroscopy, offers a robust open-microfluidic platform for label-free molecular sensing and real-time chemical/biochemical process monitoring in multi-phase systems.

Under-oil open microfluidic system (UOMS) is a rising sub-branch in open microfluidics that uses liquid-liquid boundaries rather than solid barriers to confine small volumes of liquids/analytes[1–6]. UOMS combines the advantages of single-liquid-phase open microfluidics (i.e., with fluid exposed to ambient air) and closed-chamber/channel microfluidics (i.e., with fluid confined in a closed space by solid walls) and allows for: i) flexible access to samples on a device with minimized system disturbance (e.g., evaporation and airborne contamination)[7]; ii) versatile fluid controls, including oil-water separation/emulsification[3,8], two-dimensional (2D)[9] and three-dimensional (3D)[10] fluid circuits, open-fluid particle/cell trapping, and on-demand reversible open-fluid valves[2]; iii) customizable mass transport, providing flexibility in managing gas permeation[7]; and iv) low adoption barriers and ease of implementation.

In 2020, Li et al. introduced exclusive liquid repellency (ELR) to UOMS[2], where the liquid is inherently (i.e., surface-texture and surfactant independent) and repelled (with Young's contact angle θ = 180°) on a solid surface[11,12]. ELR-empowered UOMS improves multiple critical functions in open microfluidics, including the lateral resolution of open channels from millimeter to micrometer scale, significantly increased flow rate range (e.g., can be comparable to the bloodstream in the circulatory system), open-fluid trapping of single cells and particles, and on-demand reversible open-fluid valves[2]. While the initial development of ELR-empowered UOMS focused on applications in biology and biomedicine, these critical improvements in functionality enabled by ELR lay the foundation for a broad range of applications of UOMS in chemical analysis.

[1]Department of Materials Science and Engineering, University of Wisconsin-Madison, Madison, WI 53706, USA. [2]Department of Civil and Environmental Engineering, University of Wisconsin-Madison, Madison, WI 53706, USA. [3]Carbone Cancer Center, University of Wisconsin-Madison, Madison, WI 53705, USA. [4]Department of Pathology and Laboratory Medicine, Madison, WI 53705, USA. [5]Department of Biomedical Engineering, University of Wisconsin-Madison, Madison, WI 53705, USA. ✉e-mail: cli479@wisc.edu; bu.wang@wisc.edu

Compared to closed-chamber/channel microfluidic systems, the inherent advantages of ELR-empowered UOMS mentioned above make it particularly suitable for studying multi-phase chemical reactions. Specifically, it minimizes the risk of channel clogging commonly seen in closed microfluidic systems when gas bubbles and/or solid particles are involved or generated, e.g., in multi-phase chemical reactions[13,14]. More importantly, gas permeation can be readily controlled and adjusted in UOMS by selecting the properties of the oil overlay (e.g., oil type, viscosity, thickness)[7]. Highly adjustable gas permeation through the oil overlay can be utilized to control and simulate a target gas environment of variable concentrations. In contrast, gas permeation in most closed microfluidic systems is hardly adjustable. Specifically, the solid barrier materials commonly used in closed microfluidic systems exhibit highly polarized permeability, e.g., ultra-high gas permeability with polydimethylsiloxane (PDMS) elastomer and ultra-low gas permeability with plastics such as polystyrene (PS), polymethyl methacrylate (PMMA), and polycarbonate (PC)/glass[7]. The bubble/particle clogging propensity and limited gas permeation control with closed microfluidic systems has stymied the utility of microfluidics in study of various multi-phase chemical reactions.

Furthermore, thanks to the oil overlay providing full optical access to the testing media, UOMS can be readily combined with label-free, spectroscopic detection techniques such as infrared[15-17] and Raman spectroscopy[18-20]. The combination of microfluidic platforms and in situ characterizations, in particular using IR and Raman spectroscopy, has proven to be a powerful analytical tool in many areas including biological analysis[21-24], quantitative detection[25-27], materials synthesis[28-31], real-time detection, versatility[32-36], etc. Before the

development of UOMS, Raman-integrated microfluidics were dominated by the classical closed microfluidic systems and the single-liquid-phase open microfluidic systems. Limitations of closed microfluidic systems have been outlined above. In comparison, single-liquid-phase open microfluidic systems overcome some of those limitations, but a separate set of problems exist, including system instability, e.g., media loss via evaporation and airborne contamination, and the lack of gas permeation control due to the free liquid-air interface. In comparison, UOMS, especially empowered by ELR, provides a unique solution that simultaneously overcomes issues associated with both types of microfluidics. Once integrated with spectroscopic detection, ELR-empowered UOMS could provide a robust alternative to the traditional microfluidic systems for in situ, high-throughput analysis of multi-phase chemical reactions, which would especially benefit emerging engineering fields such as photocatalysis[37-39] and $CO_2$ capture[19,40].

In spite of the advances in UOMS, previous studies of UOMS have primarily focused on device fabrication and functionalities in biology (e.g., fluidics control, cell cultivations)[2,6,7,9,10]. Applications in broader chemical analysis, especially multi-phase chemical reaction systems are critically underexplored. Additionally, in current UOMS designs, due to the low visual contrast between the substrates (e.g., glass slides) and the chemically defined surface patterns, sample loading, identification, manipulation, and characterizations under a microscope or naked eyes have been quite challenging, especially for those focal plane-sensitive tools (e.g., confocal Raman spectroscopy).

In this work, we develop visualization-enhanced UOMS (V-UOMS) and demonstrate its applications in monitoring multi-phase chemical reactions with Raman spectroscopy. Visualization enhancement of the

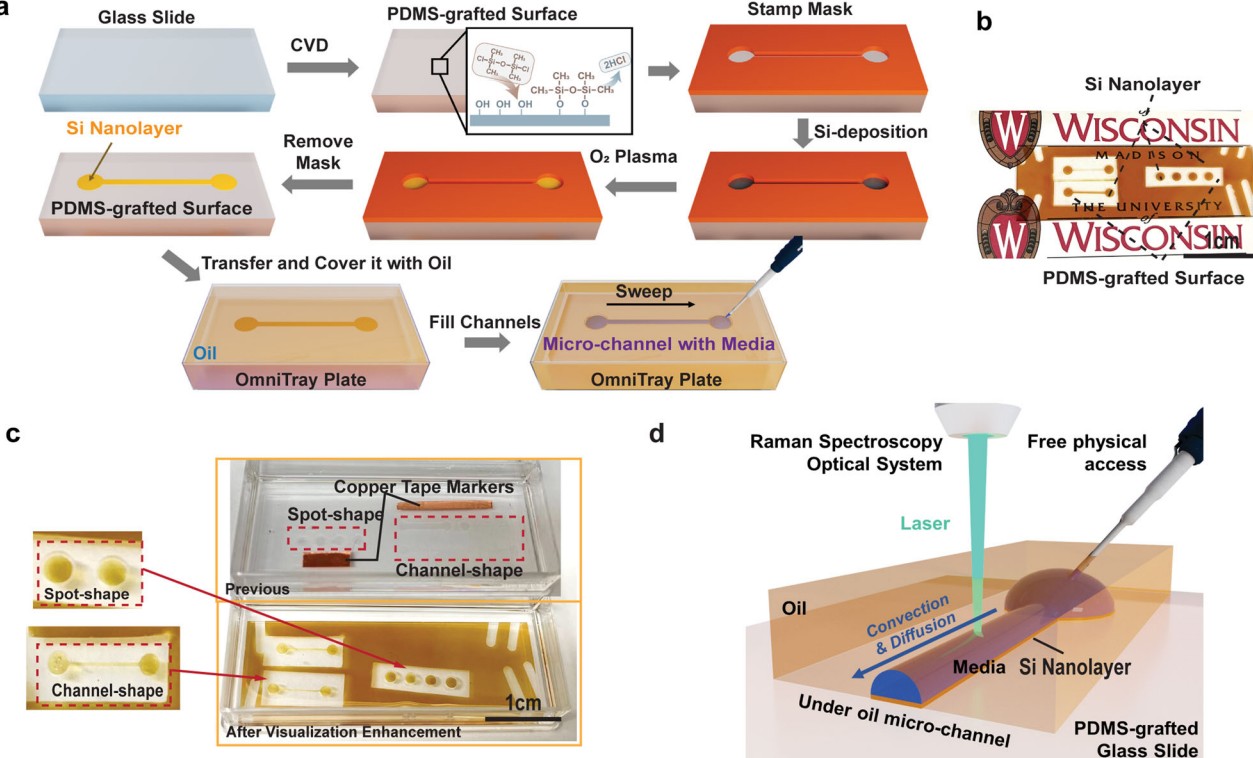

**Fig. 1 | Fabrication workflow of V-UOMS and its integration with Raman spectroscopy. a** Schematic diagram of the V-UOMS platform fabrication workflow on a glass substrate (see Methods). PDMS-silane graft via covalent bonding is first achieved on the glass slide through a chemical vapor deposition (CVD) process (see Methods). The channel patterns are then achieved through Si deposition with a press-to-seal silicone stamp mask (red). After placing the V-UOMS into an OmniTray plate (with the silicone stamp mask removed), oil (silicone oil, 5 cSt) is added to cover the glass slide. The micro-channel is then filled with aqueous media by sweeping the channel under oil with a handing drop of media using a pipette[2,12]. **b** A camera image of V-UOMS with micro-channels and micro-spots on a glass slide (3 × 1 inch) in air. The amber color is from the Si nanolayer (100 nm in thickness). **c** Camera images of UOMS (top, without visualization enhancement) and V-UOMS (bottom, with visualization enhancement) glass slides overlaid with oil in an Omnitray plate (silicone oil, 5 cSt, 4 mL per Omnitray well). Channel patterns are transparent to the naked eye without enhancement. **d** 3D schematic illustration of V-UOMS integrated with Raman spectroscopy.

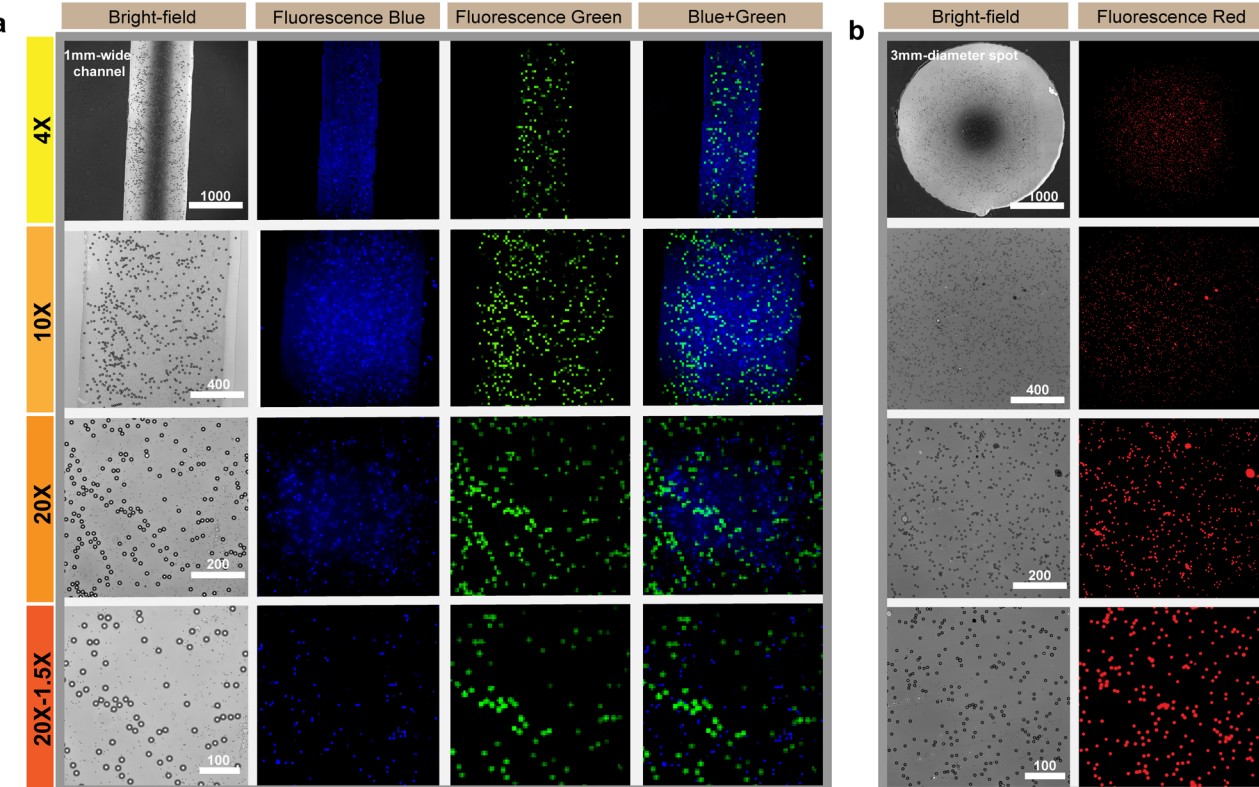

**Fig. 2 | Optical access (bright field and epifluorescence) of V-UOMS under an inverted microscope.** Microscopic images of fluorescent PS beads [blue - Ex/Em 360 nm/420 nm, 2.07 µm in diameter; green - Ex/Em 480/520 nm, 15.25 µm in diameter; far-red - Ex/Em 660/690 nm, 7.32 µm in diameter] in a 1-mm-wide micro-channel (**a**) or a 3-mm micro-spot (**b**) at different magnifications. Normalized contrast adjustment was applied throughout all the images. Scale bars: 1000 µm for 4×, 400 µm for 10×, 200 µm for 20×, and 100 µm for 30× magnification.

surface patterns is achieved by introducing a Si nanolayer (~100 nm in thickness) deposition on the patterned areas of substrates. The visual contrast between the Si nanolayer and substrates facilitates the identification of the patterned regions by naked eyes and focusing under a microscope, improving the ease and accuracy of operation. In experiments, we validate that: (i) the Si nanolayer is compatible with the ELR wettability contrast after oxygen ($O_2$) plasma treatment; (ii) the V-UOMS retains optical access with the inverted microscope setup (broadly adopted in biomedical research and labs); (iii) the Si nanolayer effectively reduces the background noise from the substrates, especially when near the media-substrate interface; and (iv) V-UOMS shows stability against different chemical (pH, hydrocarbon solvent) environments that are commonly seen in multi-phase chemical reactions. Finally, through fluid dynamics analysis and experiments, we validate V-UOMS's chemical stability and its capability to monitor gas evolution and gas-liquid reactions related to $CO_2$ capture and sequestration. V-UOMS integrated with Raman spectroscopy offers a robust open-microfluidic platform for label-free molecular sensing and real-time chemical/biochemical process monitoring in multi-phase systems.

## Results

### The under-oil microenvironment in UOMS and visualization enhancement

In UOMS, the visual contrast between the chemically defined (i.e., texture-free) surface patterns and the transparent substrate (e.g., glass) is low (Fig. 1a–c). Identifying the surface patterns, operating under naked eyes, and conducting in situ characterizations with optical microscopes can be challenging due to the limited visual contrast. To address this limitation, we utilize a Si nanolayer to prime the surface patterns, enhancing the visual contrast without compromising optical access, surface wettability, and the chemical stability/biocompatibility

of the system. Si is a commonly used reference material for Raman measurements. Unlike glass substrate that displays a broad range of Raman signals between 900 and 1100 cm$^{-1}$, Si displays a single mode at 520 cm$^{-1}$ which is routinely used for Raman calibration[41]. Previous report has revealed the noise from substrates could interfere with the Raman measurements[42]. Covering the glass substrate with a Si nanolayer offers the benefits of suppressing the substrate's noise with Raman measurements, in addition to enhancing the reflectivity of the substrate. Overall, we found that a 100 nm thickness of Si deposition effectively covers most of the glass signals in Raman measurements while retaining sufficient optical access for inverted microscopes, both of which will be discussed in detail in the following sections.

To deposit the Si nanolayer following a given surface pattern, e.g., micro-spots or micro-channels, we used an e-beam evaporator with a silicone rubber stamp mask following the workflow illustrated in Fig. 1a. After overlaid with oil (here silicone oil, 5 cSt) and filled with aqueous media [here 0.02% sodium dodecyl sulfate (SDS) aqueous solution with interfacial tension of ~40 mN/m] via under-oil sweep (Fig. 1c)[43], the V-UOMS was then tested and used with confocal Raman spectroscopy. A 3D schematic of the overall Raman V-UOMS system is shown in Fig. 1d. Compared to the original UOMS, the visibility of surface patterns is noticeably improved with the Si nanolayer (Fig. 1b, c). Surface hydrophilicity comparable to that of glass slides was achieved by treating the Si nanolayer with O2 plasma (Fig. 1a, Methods)[44]. As such, ELR-enabled operations and fluidic controls remain unaffected by visualization enhancement with the Si nanolayer (Fig. 1d).

### V-UOMS optical access (for bright-field/epifluorescence imaging on inverted microscopes) and Raman spectroscopy test

Inverted microscope setups have been used in a wide range of applications, especially in biomedical research. Keeping the Si nanolayer in

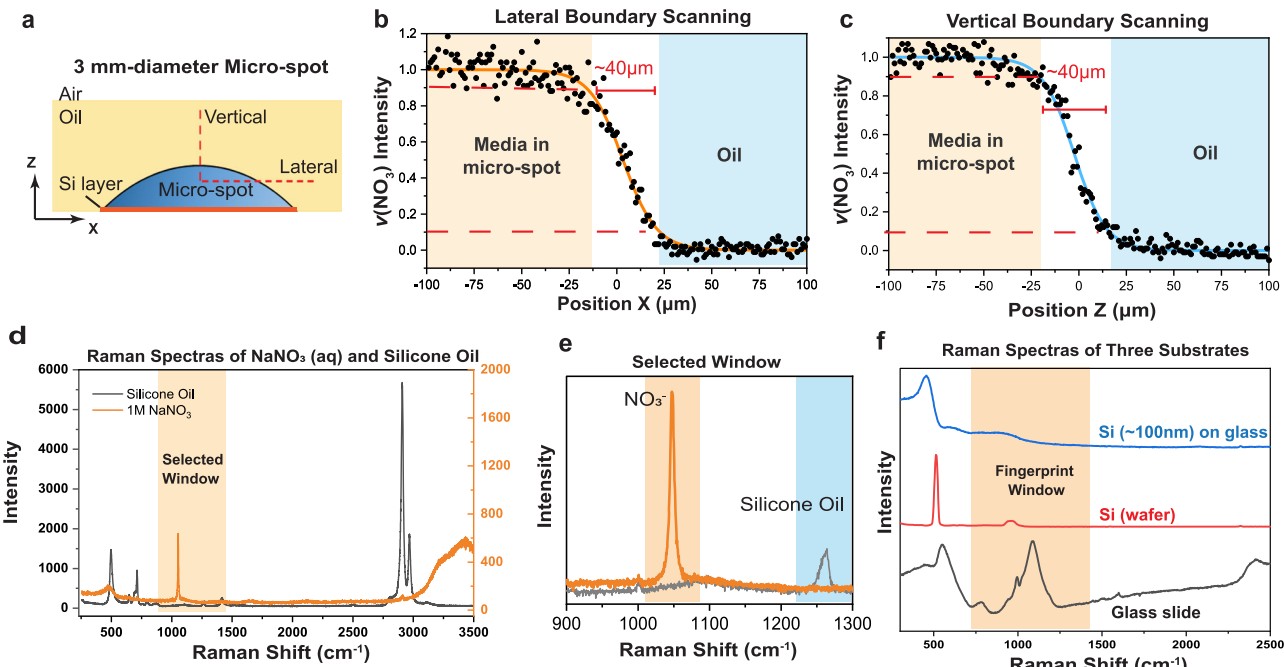

**Fig. 3 | V-UOMS Raman spectroscopy test for the boundary sharpness characterization at the oil-media interface. a** Schematic diagram showing the oil-media interfacial line scanning paths in the lateral and vertical directions. **b** Results of lateral, and (**c**) vertical scanning of the oil-media interfaces. The scattering plots are normalized based on the $\nu(NO_3^-)$ peak and then fitted with sigmoid functions. The width of the oil-media boundary is defined by 10–90% intensity. "Raman spectra of NaNO₃ (aqueous) and silicone oil": (**d**) Raman spectra of 250–3500 cm⁻¹ window and **e** 900–1300 cm⁻¹ window. **f** Comparison of the Raman spectra of three substrates, including glass slide, Si wafer, and 100 nm Si nanolayer-coated glass slide.

V-UOMS sufficiently transparent to allow uncompromised optical access to the samples through the substrate makes the system compatible with both upright and inverted microscopes. Here, we characterize a 1 mm-wide micro-channel (Fig. 2a) and a 3 mm-diameter micro-spot (Fig. 2b) in V-UOMS filled with a suspension of fluorescent PS beads under an inverted microscope (Nikon Eclipse Ti). Three types of fluorescent PS beads [blue (Excitation/Emission (Ex/Em) 360/420 nm), green (Ex/Em 480 nm/520 nm), and far red (Ex/Em 660 nm/690 nm)] were used to fill the micro-channel and micro-spot on the V-UOMS device. The system was then imaged with bright field and fluorescent channels at four different magnifications (4×, 10×, 20×, and 30×). The results (Fig. 2) show that both the tested micro-channel and micro-spot with the 100 nm Si nanolayer retain sufficient optical access. PS beads can be observed clearly under both bright fields and epifluorescence. These characterizations prove that the Si nanolayer over the substrate successfully enhances visualization without compromising the optical access through the substrate.

Next, we evaluate Raman measurements in the under-oil condition with V-UOMS, particularly at the interfaces between the media and oil/substrate. This testing is necessary to quantify and select suitable locations for characterizations with minimized interferences from the oil and/or substrate. The boundary sharpness at the oil-media-substrate interfaces can serve as an indicator to gauge the suitability of the setup for accurate and precise measurements. Here, we use an upright confocal Raman microscopy equipped with a 532 nm-wavelength laser and a 50× (Numerical Aperture, N.A. = 0.5) objective. The theoretical spatial resolution with this setting is around 0.8 μm in the lateral direction and 1.5 μm in the vertical direction[45,46].

In this testing, a 3 mm-diameter micro-spot was filled with 4 μL of 1 M NaNO₃ aqueous solution under silicone oil (5 cSt, 4 mL). We then took a vertical (z-axis) and lateral (x-axis) scanning line scanning, respectively, across the oil-media interface (Fig. 3a–c). The spectra of the NaNO₃ solution (i.e., media) and the oil are shown in Fig. 3d. The $\nu(NO_3^-)$ peak intensity in the 1020–1080 cm⁻¹ window was collected as

the signature for the NaNO₃ solution (Fig. 3e). The intensity data was normalized based on the min-max signals and then fitted with a sigmoid function (red solid line), which is shown in Fig. 3b, c. From the sigmoid function, we defined the oil-media boundary width as the distance between 10% and 90% of the signal intensity (red dashed lines in Fig. 3b, c), which was measured to be 40 μm (in both vertical and lateral scanning). This indicates that the oil phase would not affect Raman measurements at the center of the channel when the channel width or height is greater than 40 μm (20 μm from both sides). Smaller channels can also be used if the oil is chosen carefully so that its Raman signal does not mask key Raman signatures of the testing media. Similarly, a V-UOMS micro-channel cross-sectional mapping was conducted to more intuitively exhibit the oil-media boundary and Raman measurements in under-oil conditions (Fig. S1).

Further, we compared three different substrates - glass slide, 100 nm Si nanolayer-coated glass slide, and Si wafer (as the reference) - and identified that the substrate noise strongly influences the measured boundary sharpness (Fig. 3f). In the regular UOMS setup where the substrates are the glass substrate without the Si nanolayer (having a broad range of Raman signals between 700 and 2500 cm⁻¹) contributes a significant Raman noise (Fig. 3f). Moreover, the noise level from the glass substrate varies depending on the position of the Raman focal point relative to the substrate. This variability can compromise the accuracy when determining the boundary width based on signal strengths[42]. The oil-media boundary width measured in UOMS on glass substrate (without Si nanolayer), using the same method as above, are 75 μm in vertical and 150 μm in lateral scanning, which are almost 1–3 times greater than those achieved in V-UOMS for 40 μm (Fig. S3). While employing lenses with higher resolutions (such as high magnification oil-immersed lenses) and more advanced data processing techniques may be utilized to further increase the boundary sharpness in measurement, the incorporation of the Si nanolayer offers a straightforward and effective approach to reduce undesired Raman signals arising from the substrate. The Si nanolayer coating

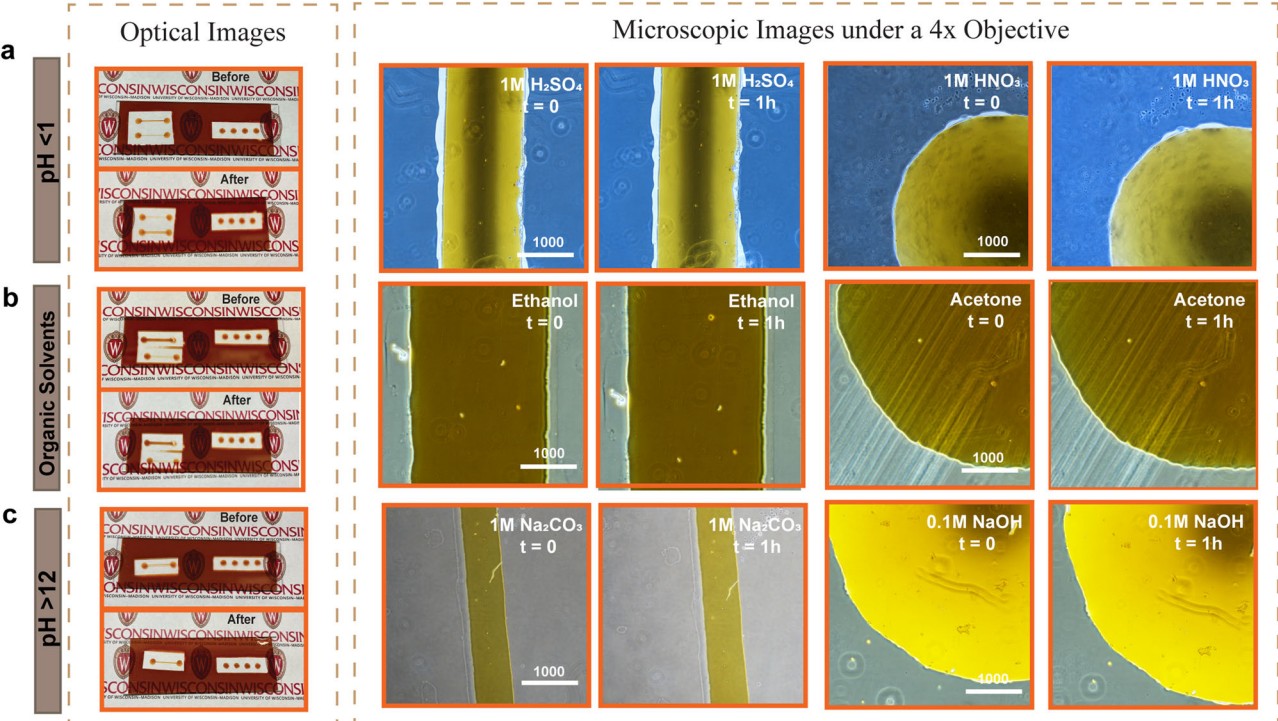

**Fig. 4 | Chemical stability of V-UOMS.** Camera and microscopic images of micro-channels and micro-spots that are filled with three types of media−(**a**) pH < 1 (1 M HNO₃ and H₂SO₄), (**b**) hydrocarbon solvents (ethanol and acetone), and (**c**) pH > 12 (1 M Na₂CO₃ and 0.1 M NaOH). The images compare the initial state and the state after 1 h, showing no significant change of the Si nanolayer in all testing conditions. (Scale bar: 1000 µm).

provides a high flexibility in selecting the substrate material when fabricating V-UOMS for Raman spectroscopy.

We noted that the width of the oil-media boundary measured in V-UOMS in the absence of strong substrate noises likely represents an interfacial layer between the oil and the aqueous media where the two phases inter-diffuse. This is determined by the nature of the oil-media interface and the temperature[47,48]. Measurements of the interfacial layer between water and kerosene oil based on optical properties have suggested the width of the water-oil boundary is in the range of 55–60 µm[49], which is close to the measured value in our system with confocal Raman (40 µm). More in-depth studies are warranted to understand the nature of the interfacial layer properties with Raman spectroscopy.

### Chemical stability of V-UOMS in different pH and hydrocarbon solvent environments

For engineering applications involving heterogeneous reactions, a broader range of pH and solvents compared to the typical biological environments (pH 6–8) is commonly used. These environments include the use of strong acids, bases, and a variety of organic solvents. To evaluate the stability of V-UOMS in such environments, here we test V-UOMS against strong acids (aqueous solution at pH < 1), strong bases (aqueous solution at pH > 12), and commonly used hydrocarbon solvents (including ethanol and acetone as examples). For acids, 1 M HNO₃, and 1 M H₂SO₄ were tested with silicone oil as the oil overlay. In the case of bases, 1 M Na₂CO₃ and 0.1 M NaOH were also examined with silicone oil. Ethanol and acetone were tested with fluorinated oil (fluorinert FC-40) due to their miscibility with silicone oil. In comparison, FC-40 comes with high chemical inertia and immiscibility with hydrocarbons. As shown in Fig. 4, three individual V-UOMS systems were prepared and tested in parallel. For each, sample media was filled in the micro-channels and micro-spots under oil. The surface patterns defined by the Si nanolayer are readily identified under a microscope

with a 4× objective. Raman spectra of sample media in three different environments are shown in Fig. S4.

In these tests, we chose a 1 h duration to evaluate the short-term stability of V-UOMS. In 1 h exposure to the testing conditions, camera images of V-UOMS confirm the overall stability of the surface patterns (Fig. 4, left). Also, no noticeable changes were observed in the morphology of the micro-channels and micro-spots defined by the Si nanolayer with bright field imaging under a microscope (Fig. 4, right). For long-term chemical stability, we recommend testing V-UOMS with the specific environment on a case-by-case basis.

### In situ Raman characterization of gas-evolution reaction in V-UOMS

Multi-phase chemical reactions involving gas and/or sediment generation are challenging for the traditional closed-chamber/channel microfluidics because gas bubbles and/or particles easily cause clogging and device failure in closed systems, especially at the sub-millimeter scale where surface tension and viscosity become dominant compared to inertial forces (e.g., flow, self-gravity)[14]. The closed-chamber/channel microfluidics use solid-physical barriers (e.g., glass, plastic) to confine the liquid media. While robust in confinement and fluid control, the solid-physical barriers are non-deformable. Once clogged with bubbles/particles, fluid functionalities are disrupted and hard to recover. By contrast, the oil-media interfaces in V-UOMS provide a well-defined but deformable liquid-liquid barrier for fluid confinement, allowing gas and/or particles to escape or settle and microchannels to spontaneously expand or contract in response to local pressure. In addition, the oil-media interfaces allow uncompromised physical access to the system thereby enabling on-device operations such as injecting reagents onto or extracting reagents from the microspots/micro-channels directly.

To demonstrate such capabilities with V-UOMS, we performed an in situ characterization of a model gas-evolution reaction between

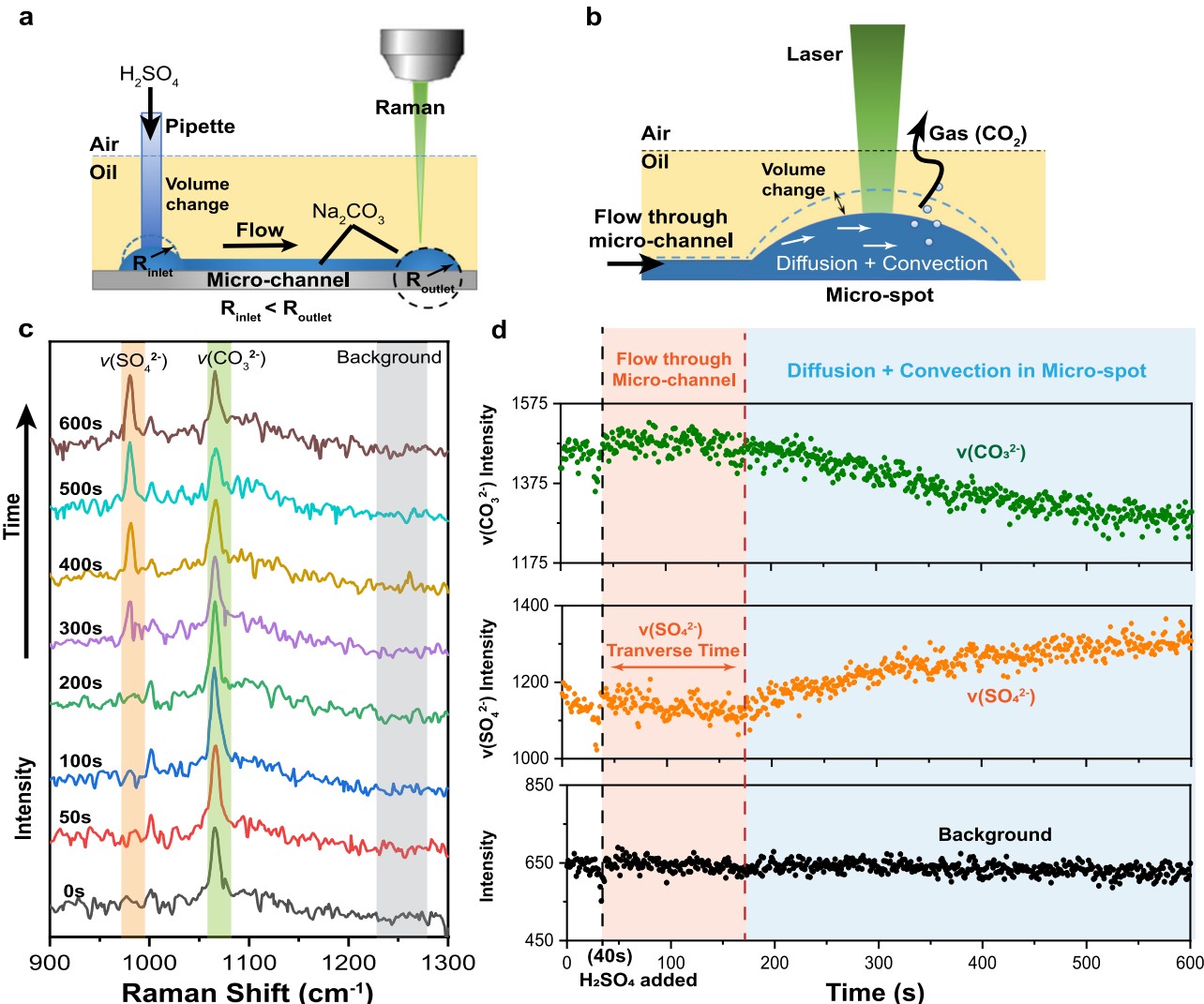

**Fig. 5 | In situ Raman characterization of gas-evolution reaction on V-UOMS.**
**a** Schematic diagram of the V-UOMS setup for a gas-evolution reaction. The initial media volume in the systems is 10 μL in total. The radius of curvature of the inlet micro-drop decreases after liquid ($H_2SO_4$) loading (4 μL). The radius of curvature differential between the micro-drops at the inlet and the outlet triggers a positive Laplace pressure drop, generating a flow from the inlet to the outlet (i.e., passive pumping)[2]. **b** Schematic illustration of the flow and diffusion at the observation end

(outlet micro-drop). The mechanism within the micro-channel is considered convection. Whereas within this micro-drop, the mechanisms governing mass transport are expected to be diffusion and convection. **c** Selected Raman spectra during the measurement and integration windows for calculating peak intensities, i.e., 960–990 $cm^{-1}$ for $v(SO_4^{2-})$, 1060–1080 $cm^{-1}$ for $v(CO_3^{2-})$, and 1220–1270 $cm^{-1}$ for background. **d** Intensity-time-series plots of the $v(CO_3^{2-})$ peak, $v(SO_4^{2-})$ peak, and background.

$Na_2CO_3$ and $H_2SO_4$. The progression of the reaction was characterized using a time-series Raman measurement. As shown in Fig. 5a, b, we use a V-UOMS (500 μm micro-channel with 3-mm diameter micro-spots on both ends) as the reactor. The V-UOMS, covered with silicone oil and filled with 10 μL 1 M $Na_2CO_3$ solution, was placed under the confocal Raman microscope with the focal point placed inside the micro-drop (formed on the micro-spot) at the observation end (see detailed discussion in SI - Passive Pumping Dynamic Flow Analysis). A 600 s time-series Raman measurement (1 scan per second) was carried out. At t = 40 s, 4 μL 1 M $H_2SO_4$ was added to the inlet micro-drop with a pipette. The liquid loading broke the pressure equilibrium, pumping liquid from the inlet micro-drop to the outlet micro-drop (Fig. 5b). As the H2SO4 solution traveled across the channel, it reacted with $Na_2CO_3$, releasing gaseous $CO_2$. The formation of gas bubbles inside the outlet micro-drop could be observed using the microscope with the naked eye. Such processes with gas evolution and fluid volume change can be hardly accommodated in closed chamber/channel microfluidic systems.

The Raman spectra at different time points are plotted in Fig. 5c, and the intensity-time plots with the Raman intensities of the $v(SO_4^{2-})$, the $v(CO_3^{2-})$, and the background are displayed in Fig. 5d. The intensity was calculated based on the peak areas highlighted in Fig. 5c. The integration windows were set as 960–990 $cm^{-1}$ for $v(SO_4^{2-})$, 1060–1080 $cm^{-1}$ for $v(CO_3^{2-})$, and 1220–1270 $cm^{-1}$ for background. As $[SO_4^{2-}]$ ions get pumped from the inlet to the outlet, the signal of $v(SO_4^{2-})$ becomes detectable at around 200 s and continues to increase afterward.

The observed time evolution of $v(CO_3^{2-})$ and $v(SO_4^{2-})$ intensity can be explained by estimating the time needed for the injected $H_2SO_4$ solution to get pumped through the micro-channel driven by passive pumping between the inlet and outlet micro-drops (see SI - Passive Pumping Dynamic Flow Analysis). Based on the calculation, it would take roughly 177 s for $H_2SO_4$ solution to reach the outlet (observation end). This estimated time agrees well with our experimental observations (Fig. 5d), where it takes approximately 130 s for the $v(SO_4^{2-})$ signal to be detected at the observation end after the $H_2SO_4$ solution

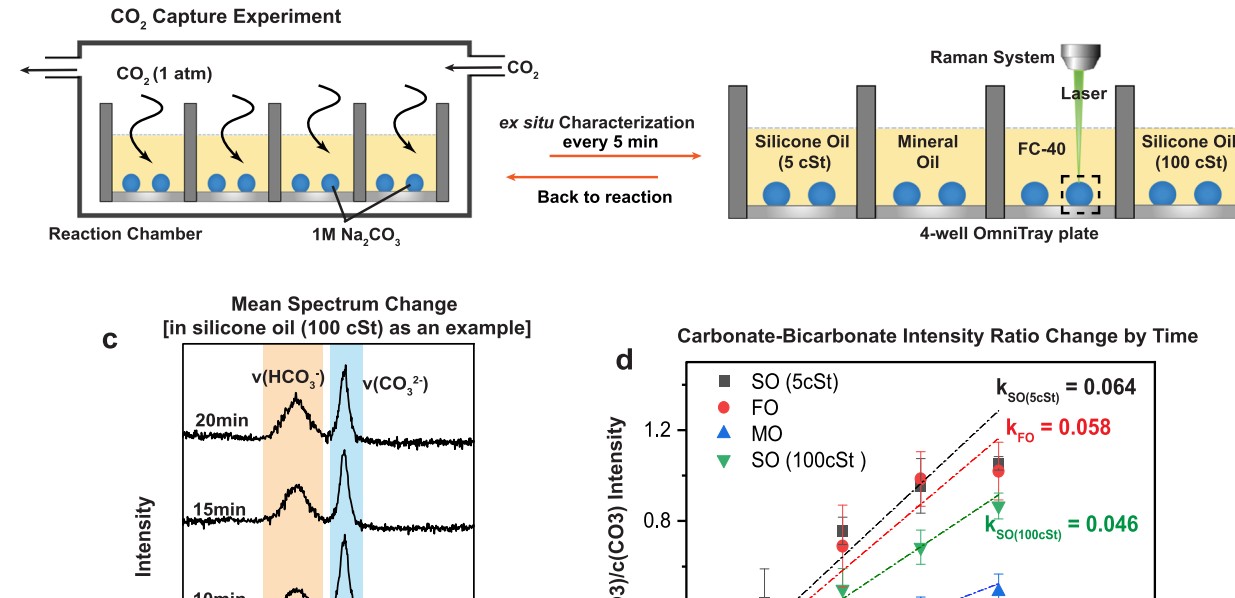

**Fig. 6 | $CO_2$ capture reaction in V-UOMS with adjustable gas permeation.**
**a**, **b** Schematic illustrating the experimental setup and workflow of $CO_2$ capture.
**a** V-UOMS with micro-spots (filled with $CO_2$ absorber $Na_2CO_3$) is kept in a 4-well OmniTray plate filled with four different types of oil [silicone oil (5 cSt), fluorinert FC-40, mineral oil, and silicone oil (100 cSt)]. **b** ex situ Raman observations are conducted at 5-min intervals. **c** Mean spectra of $\nu(HCO_3^-)$ and $\nu(CO_3^{2-})$ corresponding to the reaction from the condition of silicone oil (100 cSt), are shown as
an example. **d** Plots display the $c(HCO_3^-)/c(CO_3^{2-})$ intensity ratio during the initial 20 min of the reaction (each data point corresponds to 20 measurements). Error bars represent the 95% confidence interval. Linear regression analysis is added as dash-dot lines on the plot to show the gradient (represents reaction rate), intersecting the experimental data points. [SO: silicone oil (5 cSt and 100 cSt) at 25 °C, FO Fluorinert FC-40, MO mineral oil].

was injected at the inlet. The intensity of $\nu(SO_4^{2-})$ keeps increasing afterward, due to continued passive pumping flow and diffusion within the outlet micro-drop. Simultaneously, the intensity of $\nu(CO_3^{2-})$ reduces as $[CO_3^{2-}]$ reacts with $[H^+]$ to produce gaseous $CO_2$. During the entire experiment, the V-UOMS operation and the Raman measurement were not affected by either the volume change inside the microchannel/drop or the generation of $CO_2$ bubbles, validating a good compatibility of V-UOMS with gas evolution reactions.

### Gas-liquid reaction with adjustable gas permeation in V-UOMS
Performing heterogeneous chemical reactions in microfluidic systems, especially the gas-liquid reactions, have been drawing increasing attention recently[44,50,51]. Besides the bubble/particle clogging issue discussed above, another major limitation facing in closed-chamber/ channel microfluidics in muti-phase reaction applications is the polarized gas permeabilities of the typically used solid materials in fabricating the closed microfluidic systems. For example, PDMS elastomer (one of the most commonly used materials for fabricating microfluidics) exhibits ultra-high gas permeability while plastics (e.g., PS, PMMA, PC), and glass come with ultra-low gas permeability[7]. A reported approach to perform gas-liquid reactions in closed microfluidic systems is the introduction of gas bubbles into the channels[44,50]. However, the introduced bubbles disrupt or block the channels and mass transport. The drifting of the gas bubbles also interferes with measurements within the system[13,14].

The V-UOMS platform addresses these challenges with gas permeable oil overlay. By simply choosing oils with different gas

permeability and/or adjusting the oil layer thickness, not only gas-liquid reactions can be performed without interferences to microfluidic operations or measurements, but also the gas permeation (and consequently the effective gas partial pressure over the testing media) in each reactor unit can be individually regulated and adjusted before and during the system is running. The capability of V-UOMS to run multi-phase chemical reactions with individually and autonomously controlled gas permeation is critical for high-throughput (e.g., 384-well plate or higher capacity) mechanism study and reagent screening.

Here, we demonstrate this capability with a model gas-liquid reaction between $CO_2$ gas and $Na_2CO_3$ solution, $Na_2CO_3 + CO_2 + H_2O \rightarrow 2NaHCO_3$. This reaction reduces the $[CO_3^{2-}] / [HCO_3^-]$ ratio in the solution, which can be tracked by the signature Raman peaks of the two anions. For a short time in a solution with a given $Na_2CO_3$ concentration, the reaction rate is dictated by the $CO_2$ concentration at the oil-media interface in V-UOMS.

As shown in Fig. 6a, b, we use four types of oils [5-cSt silicone oil, fluorinert oil (FC-40), mineral oil, and 100-cSt silicone oil] to achieve different gas permeabilities. Four V-UOMS glass slides, each having 3-mm-diameter micro-spots, are respectively placed in the 4 wells of a 4-well OmniTray plate. Each V-UOMS glass slide is covered with 4 mL of one type of oil. All four micro-spots are then filled with 5 μL of 1 M $Na_2CO_3$ solution per micro-spot under oil (Fig. S5). The four-well OmniTray plate is then transferred into a $CO_2$ reaction chamber, where a constant flow of reagent grade $CO_2$ gas is introduced at 10 ccm and the pressure inside the $CO_2$ reaction chamber is maintained at 1 atm for

20 min. Every 5 min, we take the OmniTray plate out of the $CO_2$ reaction chamber and perform ex situ Raman measurements.

$[CO_3^{2-}]$ and $[HCO_3^-]$ are quantified from the Raman signal intensities of the two anions ($A_{HCO3}$ and $A_{CO3}$) using the relative molar Raman scattering factors ($J_{HCO3}$ and $J_{CO3}$):

$$\frac{cHCO_3^-}{cCO_3^{2-}} = \frac{A_{HCO3}}{A_{CO3}} \times \frac{J_{HCO3}}{J_{CO3}} \qquad (1)$$

The intensities are obtained by integrating the area under the Raman spectra between 970–1040 cm$^{-1}$ for $[HCO_3^-]$ and 1050–1075 cm$^{-1}$ for $[CO_3^{2-}]$ (see example in Fig. 6c). Baseline correction is carried out using a 4-th-order polynomial fit to the background between the 900–1200 cm$^{-1}$ window (Fig. S6). Scattering factors of 0.1667 and 0.2434 are used for the $[HCO_3^-]$ and $[CO_3^{2-}]$, respectively[46].

For each Raman measurement, we sampled 20 points within the micro-drop (X-Y scan over a 100 μm × 80 μm area) on each V-UOMS. The c($HCO_3^-$)/c($CO_3^{2-}$) ratio vs. time is plotted and fitted by linear regression in Fig. 6d, with error bars representing the 95% confidence interval. Note, the linear regression is only meant to be used as a rough indication of the short-term reaction rate; the true reaction rate needs to be determined by more sophisticated reaction-transport models. The results show that, at the same oil layer thickness, the 5-cSt silicone oil and fluorinert oil (FC-40) give the highest $CO_2$ partial pressure over the $Na_2CO_3$ solution. The 100-cSt silicone oil displays a lower reaction rate than the 5-cSt silicone oil. This is attributed to the 100-cSt silicone oil having a larger viscosity and, therefore, a lower $CO_2$ permeability. The mineral oil shows the least permeable to $CO_2$, resulting in the slowest reaction rate.

By simply changing the oil type in V-UOMS, different $CO_2$ concentrations (or effective $CO_2$ partial pressures) can be achieved over the liquid testing media, resulting in different $CO_2$ reaction rates. The gas partial pressure at the oil-media interface in V-UOMS can be predicted by multi-physics simulations based on the gas permeability and solubility of the oil in a given configuration of the device, as shown by previous studies[7]. Another way to modify the effective gas partial pressure is to change the oil layer thickness, which can be easily achieved during V-UOMS preparation or even during its operation. Taken together, V-UOMS offers a versatile microfluidic platform for studying gas-liquid reactions.

## Discussion

In this work, we demonstrate the design, validation, and utility of the V-UOMS platform for Raman monitoring of gas-liquid reactions. By simply adjusting the type and/or the thickness of the oil overlay, different gas partial pressures at the media interfaces can be achieved. This can be utilized to simulate reactions between the media and different concentrations of a gaseous regent. For instance, the capture performance of a liquid $CO_2$ absorber under different $CO_2$ concentrations (e.g., ~0.042% atmospheric to 20% flue gas concentrations) can be easily screened using a single $CO_2$ gas stream in one reaction chamber hosting multiple V-UOMS with different oil layer thicknesses. Furthermore, as demonstrated in our previous study[7], numerical simulations can be employed to quantify effective gas partial pressures for different channel geometry and V-UOMS designs. We expect that the combined use of experimental and computational approaches will provide a tool for fast analysis, optimization, and prediction of real-world, multi-phase physicochemical processes for applications such as $CO_2$ capture and conversion. In such applications, in situ characterizations can be achieved with a specifically designed gas reaction chamber having a quartz window for Raman observations.

A frequently expressed concern during the development of UOMS is the possible extraction of molecules (especially lipophilic) by the oil phase from the aqueous phase. In our previous study, we used an ultra-performance liquid chromatography-tandem mass

spectrometer (UPLC-MS) to systematically analyze possible molecule loss in UOMS cell culture[7]. The results showed high retention of lipophilic molecules in the culture media from the UOMS cell culture due to a biopolymer/surfactant layer stabilized at the oil/media interface. In the case of surfactant-free conditions, the degree of oil extraction of hydrophilic hydrocarbons from the aqueous media phase can be estimated through numerical simulation or experimental methods (e.g., IR and Raman spectroscopy). Oil extraction also highly depends on the oil type. Fluorinated oil is known for its chemical inertia and does not dissolve most hydrocarbons except for fluorinated reffents. Selecting an appropriate oil phase can minimize the extraction of lipophilic reagents.

During spectroscopy measurements, the presence of the oil phase may influence signal collection, leading to varied signal-to-noise ratio. In this work, we proved under our current settings, the oil phase did not affect Raman characterizations at the center of the channel when the channel width or height is greater than 40 μm (Fig. 3). More in-depth work is being carried out to systematically evaluate the influence of oil signals in V-UOMS and methods to mitigate them.

V-UOMS, integrating optical spectroscopies including laser confocal, multiphoton, Raman, and IR, holds promise for label-free, in situ characterization and reaction monitoring in a wide range of applications in chemical engineering, materials science and engineering, and biomedical engineering. Future studies focusing on improving characterization resolutions and data processing, optimizing microreactor designs, and demonstrating particle trapping and sorting will further broaden the platform's applications.

## Methods

### V-UOMS preparation workflow (glass slides as substrate)
A premium microscope slide (VWR 48300-026) was first treated by $O_2$ plasma (Diener Electronic Femto) at 100 W for 3 min and then moved to a vacuum desiccator (Bel-Art F420220000, Thermo Fisher Scientific, 08-594-16B) for CVD with PDMS-silane (1,3-dichlorotetramethylsiloxane; Gelest, SID 3372.0). 10 μl liquid PDMS-silane was vaporized under pumping (3 min) and deposited on glass at 60 °C for 15 min. The PDMS-grafted slide was then rinsed with ethanol (anhydrous, 99.5%) and deionized water, and dried by $N_2$ for use. The PDMS-grafted slide was then covered by a press-to-seal silicone stamp mask (from Grace Bio-Labs) through holes/channels of a certain shape and a 100 nm thick nanolayer of Si was deposited at a rate of 1 Å/s through e-beam vapor deposition (UW-Madison NFC Fabricated). The surface was finally treated with $O_2$ plasma at 250 W for 15 s before removing the masks for the next steps.

### Preparation of under-oil micro-channels and imaging under the inverted microscope
The prepared V-UOMS was put in a four-well polystyrene (PS) Omni-Tray Plate (267061, Thermo Fisher Scientific) and overlaid with silicone oil (5 cSt) (317667, Sigma-Aldrich). Pipette with a flat-tip pipette tip (02-707-134, Thermo Fisher) was used to sweep a hanging drop of the target aqueous media over the micro-channels to spontaneously distribute a volume to the patterned areas defined by the Si nanolayer. An antistatic gun (EMS 60610, Zerostat 3 Milty) was used to generate perturbation at the oil layer to facilitate the displacement of oil by the aqueous media during the sweep. Fluorescently labeled PS beads (Bangs Laboratories) were filled in the micro-channels by sweeping under oil. Imaging of the PS beads was performed on an epi-fluorescence microscope (Nikon Eclipse Ti) with bright-field and fluorescent channels for blue (Ex/Em 360 nm/420 nm), green (Ex/Em 480 nm/520 nm), and far red (Ex/Em 660 nm/690 nm).

### Raman spectroscopy on V-UOMS
Raman spectra were collected using a Horiba LabRAM HR Evolution confocal Raman microscope. For Raman measurements, the V-UOMS

was mounted on the x-y-z automated stage under the microscope. A 100 mW 532 nm laser was used as an excitation light source along with an 1800 line/mm grating spectrometer. The laser power was adjusted to 50% by a filter. A 50× long-working-distance objective with an N.A. of 0.5 was used for all measurements. For point observations, measurements were accumulated 3 times with a 5 s accumulation time. For mapping, measurements were carried out 2 times with a 1 s accumulation time. For in situ measurement, measurements were taken at a rate of 1 spectrum-per-second to reach a continuous monitor.

## Measurement of oil-media boundary sharpness and mapping of channel cross-section by Raman

Media (1 M $NaNO_3$) solution was mixed with 0.05 wt% SDS to reduce its surface tension for under-oil sweep distribution. For boundary sharpness measurement, a 3-mm diameter micro-spot was filled with 4 µL media. The sharpness measurement was carried out by a vertical (z-axis) and lateral (x-axis) scanning line scanning, respectively, across the oil-media interface (Fig. 3a–c). Similar to boundary scanning, for 3D mapping, a 10-mm long 0.5-mm wide micro-channel with two 3-mm micro-spots at both ends was filled with 10 µL media. Then, cross-section mapping was carried out in the Y-Z plane near the micro-channel mid-point (channel cross-section; see Fig. S1a). The cross-section heat map was then reconstructed based on the intensity of $v(NO_3^-)$ (1020–1080 cm$^{-1}$ window; Fig. S1b) and silicone oil (1220–1280 cm$^{-1}$ window; Fig. S1c) respectively.

## In situ Raman characterization of gas-evolution reaction in V-UOMS

In this experiment, a continuous Raman observation was conducted to capture the reactant/product signals over time. 1 M $Na_2CO_3$ and 1 M $H_2SO_4$ solutions were supplemented with 0.05 wt% SDS. 10 µL $Na_2CO_3$ solution was filled in a 10-mm long 0.5-mm wide micro-channel with two 3-mm micro-spots at both ends (Fig. 4). A 600 s time-series Raman measurement was taken at a rate of 1 spectrum-per-second continuously. At t = 40 s, 4 µL 1 M $H_2SO_4$ was added to the inlet micro-drop with a pipette.

## Gas-liquid reaction with adjustable gas diffusion in V-UOMS

The reaction was conducted in a $CO_2$ chamber. $CO_2$ gas flowed through an inlet at 10-psi pressure to maintain a ~1 atm $CO_2$ atmosphere inside the chamber. 1 M $Na_2CO_3$ aqueous with 0.05% SDS was selected as media. Then, four V-UOMS, each having a micro-spot, were placed in the four wells, respectively. All four micro-spots were filled with 5 µL media afterward (micro-spots after loaded with media: 3 mm in base length, 1.18 mm in height for 5 µL/micro-spot). The four V-UOMS were respectively covered with 4 mL of four types of oils (silicone oil with viscosities of 5 cSt and 100 cSt, fluorinert oil FC-40, and mineral oil). The distance from the top of the micro-spots to the air was approximately 1 mm. The four-well OmniTray plate was then transferred into the $CO_2$ chamber. Every 5 min, the V-UOMS were taken out of the chamber for ex situ Raman measurements. The OmniTray was then put back into the $CO_2$ reaction chamber until the next time point. The reaction was controlled for 20 min in total.

## Data availability

All other data supporting the findings of the study, including experimental procedures and characterization, are available within the paper and its Supplementary Information, or from the corresponding author upon request. Source data are provided with this paper through figshare https://doi.org/10.6084/m9.figshare.24905139.

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

## Acknowledgements

This work is supported by the National Science Foundation under Grant No. 2132022, Support Grant from the University of Wisconsin Carbone Cancer Center Cancer Center and the College of Engineering/Grainger Institute for Engineering, NIH P30CA014520, NIH R01AI54940, and NIH R01CA24749. The authors gratefully acknowledge partial support of this research by NSF through the University of Wisconsin Materials Research Science and Engineering Center (DMR-2309000).

## Author contributions

B.W. and C.L. conceived and designed the research. C.L. developed the under-oil open microfluidic platform. Q.C. performed the experiments with assistance from C.L. and H.Z.; Q.C. and C.L. performed the data analysis, interpretation, and visualization. B.W., C.L., and D.J.B. supervised the project. Q.C. and C.L. wrote the manuscript, and all authors revised it.

## Competing interests

D.J.B. holds equity in BellBrook Labs LLC, Tasso Inc. Stacks to the Future LLC, Lynx Biosciences LLC, Onexio Biosystems LLC, Flambeau Diagnostics LLC, and Salus Discovery LLC. The remaining authors declare no competing interests.
