## [Peer Review File · Nature Communications]

Visualization-enhanced Under-oil Open Microfluidic System for in situ Characterization of Multi-phase Chemical ReactionsREVIEWER COMMENTS

Reviewer #1 (Remarks to the Author):

Recommendation – Minor Revision

The work presented here displays the potential of V-UOMS in advancing the field of microfluidics and its applications in chemical reaction analysis. The manuscript can be accepted after minor revision.

1. The authors can check and correct the grammatical and punctuation errors throughout the manuscript. For example, “Visualization-enhanced Under-oil” can be written as “Visualization-Enhanced Under-Oil.”
2. Line 68 has a punctuation error. “versatility,32–36,”
3. The authors can cite relevant literature on Raman spectroscopy for reaction monitoring in microfluidic devices. Also, explain the differences and advantages of the proposed system.
4. Can the authors discuss potential applications beyond the specific examples mentioned in the abstract? How versatile is V-UOMS, and what other areas of chemistry could benefit from this technology?
5. The authors can check the grammatical errors and follow a logical structure. Clear and concise writing will enhance the readability of the paper. The conclusion part is a little bit lengthy.
6. The authors can include the Raman measurement details such as laser power, exposure time, and accumulation time in all the experiments (gas-evolution chemical reaction, gas-liquid reactions, etc.)

Reviewer #2 (Remarks to the Author):

In this paper, the authors present a visualization-enhanced under-oil open microfluidic system (V-UOMS) for application in broader chemical analysis. They introduce in situ characterization of multi-phase chemical reactions with Raman spectroscopy. The chemical stability and capability of this system are well-validated in the presented work. Of course, there are some questions and suggestions put forward as follows. In general, revisions are needed.

1. The testing and validation in this paper are plentiful, and a simple model of the pumping flow is provided. Further modeling of the data or theoretical analysis of the process may improve the scientific value.
2. The use of scientific notation of the viscosity in the Supplementary Information ($\mu = 13 \times 10^{-10}$ Ns/mm²) is confusing since 13 is larger than 10. By the way, we did not find this value in Ref. 5. If it could be obtained by some reasoning or calculating method used in Ref. 5, could you please explain in more detail?
3. Evidence, more or less, is required to support the superiority of V-UOMS compared with other microfluidic systems, as claimed in Conclusions and Discussion. Please also provide a brief discussion of the limitations of V-UOMS.

4. A more detailed description of the fabrication and operation is required in Methods for better understanding and reproduction of the work.
5. V-UOMS is, as named, an under-oil system. What is the influence of the existence of oil on the characterized chemical reaction?

inReviewer #1 (Remarks to the Author):

Recommendation – Minor Revision

The work presented here displays the potential of V-UOMS in advancing the field of microfluidics and its applications in chemical reaction analysis. The manuscript can be accepted after minor revision.

We appreciate the reviewer's acknowledgment of the significance of our work and are grateful for the valuable suggestions. Please find our responses to each point below:

1. The authors can check and correct the grammatical and punctuation errors throughout the manuscript. For example, “Visualization-enhanced Under-oil” can be written as “Visualization-Enhanced Under-Oil.”
2. Line 68 has a punctuation error. “Versatility,32–36,”

We thank the reviewer for catching the grammatical and punctuation errors. We have carefully revised the language throughout the manuscript and corrected those errors.

3. The authors can cite relevant literature on Raman spectroscopy for reaction monitoring in microfluidic devices. Also, explain the differences and advantages of the proposed system.

We have strengthened the introduction with the following paragraph, providing more background information about the application of Raman in microfluidics, as well as new functions enabled by V-UOMS.

“The combination of microfluidic platforms and *in situ* characterizations, in particular using IR and Raman spectroscopy, has proven to be a powerful analytical tool in many areas including biological analysis,^{1–4} quantitative detection,^{5–7} materials synthesis,^{8–11} real-time detection, versatility,^{12–16}, etc. Before the development of UOMS, Raman-integrated microfluidics were dominated by the classical closed microfluidic systems and the single-liquid-phase open microfluidic systems. Limitations of closed microfluidic systems have been outlined above. In comparison, single-liquid-phase open microfluidic systems overcome some of those limitations, but a separate set of problems exist, including system instability, e.g., media loss via evaporation and airborne contamination, and the lack of gas permeation control due to the free liquid-air interface. In comparison, UOMS, especially empowered by ELR, provides a unique solution that simultaneously overcomes issues associated with both types of microfluidics. Once integrated with spectroscopic detection, ELR-empowered UOMS could provide a robust alternative to the traditional microfluidic systems for *in situ*, high-throughput analysis of multi-phase chemical reactions, which would especially benefit emerging engineering fields such as photocatalysis^{17–19} and CO₂ capture.^{20,21”}

4. Can the authors discuss potential applications beyond the specific examples mentioned in the abstract? How versatile is V-UOMS, and what other areas of chemistry could benefit from this technology?

We thank the reviewer for this question. We have added more information regarding the potential applications of Raman and V-UOMS in the introduction, as discussed above in our response to Comment #3.

In summary, Raman has been broadly used in microfluidic systems for *in situ*, label-free reagent detection and monitoring of process/reaction kinetics. V-UOMS holds intrinsic advantages over other closed microfluidics and offers inherent advantages including clogging-free flow channels, flexible access to samples, and adjustable gas permeation. This means a broad range of chemical analysis applications that involve multi-phase reactions could benefit from V-UOMS. Examples include studies of catalytic reactions (gas-liquid-solid, liquid-solid), carbon capture (gas-liquid-solid, liquid-solid), synthesis processes for MOFs, 2D materials, and organic compounds that involve phase separation, etc. With its adjustable gas

permeation and clogging-free flow channels, it could also play an important role in bio-reactions where gas exchange is crucial. Overall, while the abstract focuses on specific examples, we believe that the versatility of V-UOMS holds promise for broad applicability across various domains of chemistry.

5. The authors can check the grammatical errors and follow a logical structure. Clear and concise writing will enhance the readability of the paper. The conclusion part is a little bit lengthy.

We have carefully revised the writing throughout the manuscript. Specifically in the conclusion section, we removed the repetitive content (the first paragraph in the original manuscript) and added a paragraph to discuss the limitations of the current V-UOMS, potential enhancement, and further research directions (which is also suggested by Reviewer #2 - Comment #5).

6. The authors can include the Raman measurement details such as laser power, exposure time, and accumulation time in all the experiments (gas-evolution chemical reaction, gas-liquid reactions, etc.)

We thank the reviewer for the suggestion. More details of the Raman setup, including laser power, exposure time, and accumulation time for Raman measurement, have been added to the Methods.

Reviewer #2 (Remarks to the Author):

In this paper, the authors present a visualization-enhanced under-oil open microfluidic system (V-UOMS) for application in broader chemical analysis. They introduce in situ characterization of multi-phase chemical reactions with Raman spectroscopy. The chemical stability and capability of this system are well-validated in the presented work. Of course, there are some questions and suggestions put forward as follows. In general, revisions are needed.

We appreciate the reviewer's positive acknowledgment of our work and the recommendations for enhancing the manuscript. Our response to each comment is provided below:

1. The testing and validation in this paper are plentiful, and a simple model of the pumping flow is provided. Further modeling of the data or theoretical analysis of the process may improve the scientific value.

We appreciate the reviewer's suggestion regarding further modeling of flow kinetics. The analytical model in the manuscript for the passive pumping-driven lateral flow was adapted from previously published works. (*G. M. Walker, et al., Lab on a chip, 2002*) Passive pumping is a well-studied pumping mechanism defined by a pressure (e.g., Laplace pressure and/or hydrostatic pressure) difference across a microfluidic channel.

In response to the reviewer's comment, we have refined the section on Passive Pumping Dynamic Flow Analysis and the corresponding figures in SI to further articulate the fluid operation and parameter analysis in each of the four stages in the gas-evolution experiment. They includes Stage (i) – Adding media by under-oil sweep, Stage (ii) - Reaching pressure equilibrium, Stage (iii) - Adding reagent to the inlet spot, and Stage (iv) – Passive pumping process towards new equilibrium, which is now modeled using a finite difference calculation. In future studies, we plan to introduce more sophisticated numerical modeling (e.g., COMSOL Multiphysics) to predict and quantify the flow dynamics for a given experimental setup. This approach can also be used to study gas permeation through the oil layer, as demonstrated in our previous publication. (*Li et al., Adv. Sci., 2022*)

2. The use of scientific notation of the viscosity in the Supplementary Information ($\mu = 13 \times 10^{-10}$ Ns/mm²) is confusing since 13 is larger than 10. By the way, we did not find this value in Ref. 5. If it could be obtained by some reasoning or calculating method used in Ref. 5, could you please explain in more detail?

We thank the reviewer for catching the issues. The scientific notation has been corrected in the manuscript. As for Ref. 5 in SI, it was a citation error. We apologize for this oversight. The correct reference is (*R.J. Correia, et al., J. Chem. Eng. Data 1980*), which has been updated in the SI.

3. Evidence, more or less, is required to support the superiority of V-UOMS compared with other microfluidic systems, as claimed in Conclusions and Discussion. Please also provide a brief discussion of the limitations of V-UOMS.

The main superiorities of V-UOMS over closed systems that we aimed to demonstrate in this study include (i) free physical access to samples with minimized system disturbance, (ii) clogging-free flow channels during gas evolution reactions, and (iii) adjustable gas permeation. Experimental evidences for those capabilities are provided in Sections *In situ Raman characterization of gas-evolution reaction in V-UOMS*, and *Gas-liquid reaction with adjustable gas permeation in V-UOMS*. As discussed in the introduction (paragraphs 3-4), these capabilities are difficult to achieve with closed microfluidics or single-liquid-phase open microfluidic systems that have traditionally dominated microfluidics-based in situ characterizations using laser spectroscopic methods.

The discussion of limitations and the ongoing/future work have been added to the manuscript. Please see details in our response to Comment #5.

4. A more detailed description of the fabrication and operation is required in Methods for better understanding and reproduction of the work.

We thank the reviewer for the suggestion. More information about experimental settings has been added to Methods. Those revisions include:

- A. More details were added regarding the settings for Raman measurements (e.g., laser power).
- B. Fabrication and operation details were added for a better understanding of the procedure.
- C. Details were updated about device types/names used in this work.

We hope these additions offer a clearer understanding of our experimental approach.

5. V-UOMS is, as named, an under-oil system. What is the influence of the existence of oil on the characterized chemical reaction?

As the reviewer alluded to, the presence of oil can affect the chemical reaction in some systems through the extraction of lipophilic reactants/products. The oil layer can also influence the signal collection during spectroscopic characterizations. We appreciate the reviewer for pointing out those issues, as they were not discussed in the original manuscript. We have now added the following discussion in the revised manuscript to clarify those issues.

“A frequently expressed concern during the development of UOMS is the possible extraction of molecules (especially lipophilic) by the oil phase from the aqueous phase. In our previous study, we used an ultra-performance liquid chromatography-tandem mass spectrometer (UPLC-MS) to systematically analyze possible molecule loss in UOMS cell culture.²³ The results showed high retention of lipophilic molecules in the culture media from the UOMS cell culture due to a biopolymer/surfactant layer stabilized at the oil/media interface. In the case of surfactant-free conditions, the degree of oil extraction of hydrophilic hydrocarbons from the aqueous media phase can be estimated through numerical simulation or experimental methods (e.g., IR and Raman spectroscopy). Oil extraction also highly depends on the oil type. Fluorinated oil is known for its chemical inertia and does not dissolve most hydrocarbons except for fluorinated reagents. Selecting an appropriate oil phase can minimize the extraction of lipophilic reagents.

During spectroscopy measurements, the presence of the oil phase may influence signal collection, leading to a varied signal-to-noise ratio. In this work, we proved under our current settings, the oil phase did not affect Raman characterizations at the center of the channel when the channel width or height is greater than 40 μm (Fig. 3). More in-depth work is being carried out to systematically evaluate the influence of oil signals in V-UOMS and methods to mitigate them.”

REVIEWERS' COMMENTS

Reviewer #1 (Remarks to the Author):

The manuscript can be accepted for publication. Still the conclusion part is lengthy and not concise.

Reviewer #2 (Remarks to the Author):

The authors have answered all my concerns. I recommend this manuscript.